

# Metabolic control in patients with type 2 diabetes mellitus in a public hospital in Peru: a cross-sectional study in a low-middle income country

Irma Elizabeth Huayanay-Espinoza[1], Felix Guerra-Castañon[1], María Lazo-Porras[1,2], Ana Castaneda-Guarderas[1,2,3,4], Nimmy Josephine Thomas[5], Ana-Lucia Garcia-Guarniz[1], Augusto A. Valdivia-Bustamante[1] and Germán Málaga[1,2,6,7]

[1] Unidad de Conocimiento y Evidencia (CONEVID), Universidad Peruana Cayetano Heredia, Lima, Peru
[2] CRONICAS Centre of Excellence in Chronic Diseases, Universidad Peruana Cayetano Heredia, Lima, Peru
[3] Knowledge and Evaluation Research (KER) Unit, Mayo Clinic, Rochester, MN, United States
[4] Emergency Medicine, Mayo Clinic, Rochester, MN, United States
[5] Rollins School of Public Health, Emory University, Atlanta, GA, United States
[6] Faculty of Medicine ''Alberto Hurtado,'' Universidad Peruana Cayetano Heredia, Lima, Peru
[7] Department of Internal Medicine, Hospital Cayetano Heredia, Lima, Peru

Corresponding author
Irma Elizabeth Huayanay-Espinoza,
irma.huayanay@upch.pe

## ABSTRACT

**Objective**. The objective of this study was to assess patients' achievement of ADA (American Diabetes Association) guideline recommendations for glycosylated hemoglobin, lipid profile, and blood pressure in a type 2 diabetes mellitus (T2DM) outpatient clinic in a low-middle income country (LMIC) setting.

**Methods**. This is a descriptive cross-sectional study with 123 ambulatory T2DM patients who are being treated at a public hospital in Lima, Peru. Data was gathered via standardized interviews, clinical surveys, and anthropomorphic measurements for each patient. Blood samples were drawn in fasting state for measures of glucose, glycosylated hemoglobin (HbA1c), and lipid profile. Laboratory parameters and blood pressure were evaluated according to ADA recommendations.

**Results**. Of the 123 patients, 81 were women and the mean age was 61.8 years. Glycemic control was abnormal in 82 (68.33%) participants, and 45 (37.50%) were unable to control their blood pressure. Lipid profile was abnormal in 73 (60.83%) participants. Only nine (7.50%) participants fulfilled ADA recommendations for glycemic, blood pressure, and lipid control.

**Conclusions**. Amongst individuals with type 2 diabetes, there was poor attainment of the ADA recommendations (HbA1c, blood pressure and LDL-cholesterol) for ambulatory T2DM patients. Interventions are urgently needed in order to prevent long-term diabetic complications.

## INTRODUCTION

Type 2 Diabetes Mellitus (T2DM) is a chronic disease with a growing prevalence world-wide (*Fauci et al., 2009*; *Aschner et al., 2014*). In 2014, the International Diabetes Federation (IDF) estimated that the disease prevalence in South/Central America and Peru was 8% and 6.1%, respectively (*Aschner et al., 2014*) . However, *Seclen et al. (2015)* found a slightly higher national prevalence of 7% in Peru and 8.4% in metropolitan Lima—a prevalence that has almost doubled in the last seven years.

Metabolic control in T2DM is a critical component in diabetes care. Without well-established metabolic control, complications can arise increasing mortality rates and lowering quality of life—this represents an important burden of disease for low-middle income countries (LMICs) (*Aschner et al., 2014*; *Huang et al., 2014*). Additionally, several comorbidities are related to poor metabolic control including dyslipidemia, hypertension, and obesity, increasing the risk of long-term macro and micro-vascular complications (*Colosia, Palencia & Khan, 2013*). It has been reported that more than 80% of deaths associated with T2DM in LMICs occur due to poor metabolic control (*World health Organization, 2014*). A multicenter study performed in the US comparing glycemic control in Hispanic/Latino, non-Hispanic white, and non-Hispanic black populations found that the non-Hispanic black and white populations had better glycemic control when compared to their Hispanic/Latino counterparts (*Schneiderman et al., 2014*). They reported that Hispanic/Latino populations had the lowest percentage of participants with good glycemic control among the three groups, 47.3% compared to 52.9% and 52.6% in non-Hispanic white and black populations, respectively. In another multicenter cross-sectional study conducted in nine countries in Latin America (*Lopez Stewart et al., 2007*) ,the overall poor glycemic control (HbA1c $\geq$ 7% (53 mmol/mol)) was 56.8% while Peru had the worst numbers among all with 70% having poor glycemic control. Another study conducted within an elderly population in Costa Rica reported that 37% had poor metabolic control (HbA1c $\geq$ 7% (53 mmol/mol)), 78% had a systolic blood pressure $\geq$ 130 mmHg, 66% had a diastolic blood pressure $\geq$ 80 mmHg, and 78% had a LDL-cholesterol $\geq$ 100 mg/dl (*Brenes-Camacho & Rosero-Bixby, 2008*).

Long term care of T2DM patients represents a grand challenge for health care systems around the world (*Aschner et al., 2014*). In order to evaluate the quality of T2DM management, it has been widely agreed that the proper management and control of glycosylated hemoglobin (HbA1c), LDL-cholesterol and blood pressure are the key to prevent complications (*American diabetes Association, 2016*). These can also be used to measure the quality of diabetic healthcare in different health systems (*American diabetes Association, 2016*). Moreover, it is well-known that factors associated with a healthy lifestyle (*Westman et al., 2008*), regular physical activity (three times a week), and adherence to treatment can positively influence the course of the disease (*American diabetes Association, 2016*). There are several factors such as ethnicity and depression that have not been completely explored. With regards to depression, some studies have found a 15–20% correlation between T2DM and depression incidence,

although a significant association has not yet been found (*Gonzales et al., 2007*; *Zuberi, Syed & Bhatti, 2011*; *Stanković, 2011*).

In South America, particularly in Peru, there are few studies that evaluate demographic and clinical characteristics associated with glycemic control, blood pressure, and lipid profile in a clinical setting (*Stanković, 2011*). Such a study could help to identify the main factors that affect the course of T2DM, thus letting physicians and patients work on modifying strategies to improve the patient's prognosis.

The objectives of this study were to: (1) explore the quality of T2DM control through achievement of the American Diabetes Association (ADA) recommendations, and (2) determine whether demographic characteristics, lifestyle choices, and clinical parameters of patients in a public hospital in Peru have an impact on the quality of control of T2DM.

## MATERIALS AND METHODS

### Study design and participants

We conducted a cross-sectional study in which participants diagnosed with T2DM were recruited between March and July 2012 from the Endocrinology clinic of Hospital Cayetano Heredia in Lima, Peru. This is a university hospital located in San Martin de Porres, a low-income district north of the Peruvian capital (*Acevedo, Cisneros & Curaca, 2014*).

The recruitment strategy was convenience sampling, with patients recruited while they were waiting for their routine clinical encounter. The inclusion criteria included a diagnosis of T2DM for more than 12 months from the recruitment day, ≥18 years old, and at least one visit to the Endocrinology clinic in the previous year. Patients were excluded if they were diagnosed with secondary or gestational diabetes, chronic non-cardio-metabolic diseases (systemic lupus erythematous, rheumatoid arthritis, chronic obstructive pulmonary disease), or any mental illness or incapability. Patients who had presented with major complications of T2DM (stroke, coronary heart disease, hyperglycemic/hypoglycemic crisis, diabetic foot) in the previous year and those who had been hospitalized within the previous six months were also excluded due to the decline in social, physical and psychological functionality after these major complications and their added risk factors for mortality (*Simpson & Pilote, 2005*; *Resnick et al., 2004*).

### Assessment and outcome measures

The participants were assessed using a standardized interview and an evaluation of their medical records, using the International Physical Activity Questionnaire (IPAQ), the Scale of Adherence to Diabetes Mellitus Type 2 Treatment (Escala de Adherencia al Tratamiento de la Diabetes Mellitus tipo II—EATDM-III), and Center for Epidemiological Studies-Depression (CES-D) scale (*Handelsman et al., 2015*; *IPAQ Research Committee, 2015*; *Villalobos-Pérez et al., 2006*). For laboratory measurements, a venous sample of 10cc was taken from the arm in the morning after a minimum 8 h fast from which fasting glucose, glycosylated hemoglobin, and lipid profile were measured.

To define "poor metabolic control," the American Diabetes Association (ADA) recommendations were used: poor glycemic control (HbA1c $\geq$ 7% (53 mmol/mol)), poor controlled blood pressure (BP $\geq$ 140/90 mmHg), and poor controlled LDL-cholesterol (LDL $\geq$ 100 mg/dL) (*American diabetes Association, 2016*). Also, we include an additional analysis using HbA1c $\geq$8%, because some studies have recommended individualized treatment for each patient and propose the use of a less stringent parameter (*American diabetes Association, 2016*; *Handelsman et al., 2015*).

Demographic variables were also collected including gender, age (<65 years old or $\geq$65 years old), and marital status (in a couple: married, cohabiting, or single: not in a relationship, widowed, divorced). The clinical data collected included: years of disease (1–10 years or $\geq$10 years); treatment (none, oral anti-diabetic, insulin, both); body mass index (BMI), and chronic diabetic complications defined as microvascular (retinopathy, nephropathy, and neuropathy) and macrovascular (coronary heart disease, stroke, and peripheral artery disease).

Participant's activity levels were assessed using the IPAQ with three categories: inactive, minimally active, and HEPA active (health enhancing physical activity) (*IPAQ Research Committee, 2015*) (Table S1).

The scale used to measure adherence was EATDM-III, which has previously been used in Costa Rica (*Villalobos-Pérez et al., 2006*). This scale has seven categories including family support, communal organization, physical exercise, medical control, hygiene and self-care, diet, and appreciation of the physical condition. The questionnaire has 55 questions and each one has a score between 0 and 4. The total obtained is divided by the maximum score points (220 points) and then multiplied by 100%. Scores near 100% signify more adherence (*Villalobos-Pérez et al., 2006*).

Depression was evaluated using the CES-D. A score greater than or equal to 24 is considered positive for the presence of depression. The scale was previously validated for the Peruvian population with a sensitivity of 91.4% and specificity of 96.7% (*Ruiz-Grosso et al., 2012*).

## Sample size and calculation

The sample size of 123 participants was based on the DEAL study (Diabetes en America Latina) (*Lopez Stewart et al., 2007*), using a prevalence of poor glycemic control of 56.7% with a confidence level of 95% ($Z = 1.96$). Calculated power for depression was 28% (*Crispin, Robles & Bernabe, 2015*).

## Statistical analysis

Statistical analyses were performed using Stata 12.0. Demographic variables (age, gender, marital status), clinical criteria (disease duration, treatment, complications), lifestyle factors (IPAQ, EATDM-III), and comorbidity (depression) variables were compared against glycosylated hemoglobin looking for significant associations. Chi-square was used for comparison of qualitative variables and $t$-test for comparison of a quantitative vs qualitative variable.

Multivariate models were generated using Poisson regression reporting prevalence ratios (PR) and 95% Confidence intervals (CI). Crude and adjusted analyses were performed for the three models generated to evaluate the independent association between (i) glycemic control, (ii) blood pressure, and (iii) LDL-cholesterol with demographic, clinical, lifestyle, and comorbidity variables.

## Ethics

The study protocol was approved by the Ethical Review Board of the Universidad Peruana Cayetano Heredia and Ethical Committee of Hospital Cayetano Heredia. Written informed consent was obtained from all patients.

## RESULTS

The recruitment rate was 89.78% (123/137). Of the 123 patients participating in the study, blood samples were received from 120 patients. Most participants (65.85%) were female and the mean age was $61.84 \pm 11.10$. In addition, majority of the patients (63.41%) were in a relationship (i.e., married, living together). Mean HbA1c was $8.2\% \pm 2.19$ (66 mmol/mol), mean LDL (mg/dL) was $107.61 \pm 39.66$ and mean blood pressure was $133.26 \pm 19.03/ 75.64 \pm 10.84$ mmHg. The prevalence of uncontrolled HbA1c was 68.33%, uncontrolled LDL-cholesterol was 60.83%, and uncontrolled blood pressure was 37.50%. With regards to comorbidities, 37.40% had hypertension and 33.33% had depression. Other characteristics of the population studied are shown in Table 1. Using a cutoff point of HbA1c of 7%, only 9 (7.5%) patients met all the standards of the ADA recommendations for T2DM, but when a cutoff point HbA1c of 8% was utilized, 17 patients met all standards (14.2%).

The patients diagnosed with diabetes for 1–10 years had better glycemic control compared to those who had had a diabetes diagnosis for $\geq 10$ years (42.37% vs. 21.31%, $p = 0.013$). Good glycemic control was present in 21.92% patients younger than 65 years old and in 46.81% of those older than 65 years ($p = 0.004$). Those patients who were not on pharmacological treatment had better glycemic control than the patients who required pharmacologic therapy ($p < 0.001$). A detailed comparison between each variable and glycemic control is shown in Table 2.

In the adjusted model, a negative association with poor glycemic control (HbA1c $\geq 7\%$) was found in patients who were 65 years or older when compared to younger participants ( PR = 0.59 ; 95% CI [0.44–0.78]) and with patients who were in a relationship compared to single patients ( PR = 0.74 ; 95% CI [0.59–0.92]). Also, a positive association with poor glycemic control was found with more than 10 years of disease (PR = 1.43 ; 95% CI [1.05–1.71]), oral antidiabetic (OAD) use plus insulin treatment (PR = 2.57 ; 95% CI [1.05–6.32]), and minimal physical activity ( PR = 1.63 , 95% CI [1.23–2.15]) in the crude and adjusted model in comparison with their counterparts. BMI was included in the adjusted Poisson regression model, but there was no association found with glycemic control, LDL-cholesterol, or blood pressure. We did not find an association between LDL-cholesterol or blood pressure and demographic, clinical, lifestyle (IPAQ, EATDM-III) and comorbidity variables (Table 3).

**Table 1  Characteristics of the study population (N = 123).**

| Variables | N (%) |
|---|---|
| **Age mean ± SD** | 61.84 (±11.10) |
| **Age groups (%)** | |
| Age < 65 years | 73 (59.35%) |
| Age ≥ 65 years | 50 (40.65%) |
| **Gender (%)** | |
| Female | 81 (65.85%) |
| Male | 42 (34.15%) |
| **Marital Status (%)** | |
| Single | 19 (15.45%) |
| Married | 53 (49.09%) |
| Living together | 25 (20.33%) |
| Divorced | 7 (5.69%) |
| Widowed | 19 (15.45%) |
| **Years of disease (%)** | |
| 1–10 years | 61 (49.59%) |
| ≥10 years | 62 (50.41%) |
| **Treatment (%)** | |
| No pharmacological treatment | 10 (8.13%) |
| OAD | 74 (60.16%) |
| Insulin | 8 (6.50%) |
| OAD plus insulin | 31 (25.20%) |
| **HbA1c (%)** | 8.23 ± 2.19 |
| **HbA1c (%) by ranges** | |
| HbA1c < 7% | 43 (35.83%) |
| HbA1c = 7–8% | 23 (19.17%) |
| HbA1c = 8–9% | 22 (18.33%) |
| HbA1c = 9–10% | 9 (7.5%) |
| HbA1c > 10% | 23 (19.17%) |
| **LDL(mg/dL), mean ± SD** | 107.61 ± 39.66 |
| **Blood pressure (mmHg), mean (%)** | |
| Without hypertension | 77 (62.60%) |
| With hypertension | 46 (37.40%) |
| **BMI, mean ± SD** | 28.5 ± 4.71 |
| **IPAQ** | |
| HEPA active | 67 (56.30%) |
| Minimally active | 25 (21.01%) |
| Inactive | 27 (22.69%) |
| **EATDM-III** | 56.69 ± 11.28 |
| **Depression** | |
| Yes | 41 (33.33%) |
| No | 82 (66.67%) |

**Notes.**

HbA1c, Glycosylated hemoglobin; BMI, Body mass index; IPAQ, International Physical Activity Questionnaire; OAD, Oral antidiabetics; EATDM-III, Escala de adherencia al tratamiento de la diabetes mellitus tipo II.

**Table 2** Characteristics of the study population according to glycemic control.

| | Good glycemic control (*n* = 38) | Poor glycemic control (*n* = 82) | *p*- value |
|---|---|---|---|
| **Demographics** | | | |
| **Age** ± SD | 64.80 ± 10.4 | 60.47 ± 11.2 | |
| **Age per range(%)** | | | **0.004** |
| Age < 65 years | 16 (21.92%) | 57 (78.08%) | |
| Age ≥ 65 years | 22 (46.81%) | 25 (53.19%) | |
| **Gender (%)** | | | 0.674 |
| Female | 24 (30.38%) | 55 (69.62%) | |
| Male | 14 (34.15%) | 27 (65.85%) | |
| **Marital status (%)** | | | 0.139 |
| Without couple | 10 (23.26%) | 33 (76.74%) | |
| With couple | 28 (36.36%) | 49 (63.64%) | |
| **Clinical** | | | |
| **BMI** ± SD | 27.7 ± 4.52 | 28.87 ± 4.78 | |
| **Years of disease (%)** | | | **0.013** |
| 1–10 years | 25 (42.37%) | 34 (57.63%) | |
| ≥10 years | 13 (21.31%) | 48 (78.59%) | |
| **Treatment(%)** | | | **<0.001** |
| No pharmacological treatment | 7 (70%) | 3 (30%) | |
| OAD | 28 (38.89%) | 44 (61.11%) | |
| Insulin | 2 (25%) | 6 (75%) | |
| OAD plus Insulin | 1 (3.33%) | 29 (96.67%) | |
| **Complications (%)** | | | |
| Microvascular | | | 0.593 |
| Yes | 12 (28.57%) | 30 (71.43%) | |
| No | 26 (33.33%) | 52 (66.67%) | |
| Macrovascular | | | 0.597 |
| Yes | 7 (36.84%) | 12 (63.16%) | |
| No | 31 (30.69%) | 70 (69.319%) | |
| **Lifestyle** | | | |
| **IPAQ (%)** | | | 0.143 |
| HEPA active | 25 (37.31%) | 42 (62.69%) | |
| Minimally active | 4 (16%) | 21 (84%) | |
| Inactive | 8 (29.63%) | 19 (70.37%) | |
| **EATDM-III** ± SD | 57.26 ± 11.81 | 56.42 ± 11.10 | |
| **Comorbidities** | | | |
| **Depression (%)** | | | 0.684 |
| Yes | 12 (29.27%) | 29 (70.73%) | |
| No | 26 (32.91%) | 53 (67.09%) | |

Notes.

BMI, Body mass index; IPAQ, International Physical Activity Questionnaire; OAD, Oral antidiabetics; EATDM-III, Escala de adherencia al tratamiento de la diabetes mellitus tipo II.

**Table 3 Factors associated with glycemic control, LDL cholesterol and blood pressure.**

| | Glycemic control | | LDL cholesterol | | Blood pressure | |
|---|---|---|---|---|---|---|
| | Crude model PR (95% IC) | Multivariable model[a] PR(95% IC) | Crude model PR (95% IC) | Multivariable model[a] PR ( 95% IC) | Crude model PR (95% IC) | Multivariable model[b] PR (95% IC) |
| **Gender** | | | | | | |
| Female | 1 (Reference) | 1 (Reference) | 1 (Reference) | 1 (Reference) | 1 (Reference) | 1 (Reference) |
| Male | 0.95 (0.73–1.23) | 1.07 (0.84–1.36) | 1.00 (0.74–1.36) | 1.01 (0.72–1.42) | 1.06 (0.66–1.72) | 1.01 (0.62–1.64) |
| **Age** | | | | | | |
| Age < 65 years | 1 (Reference) | 1 (Reference) | 1 (Reference) | 1 (Reference) | 1 (Reference) | 1 (Reference) |
| Age ≥ 65 years | **0.68(0.51–0.92)** | **0.59(0.44–0.78)** | 0.91 (0.67–1.24) | 0.98 (0.70–1.38) | 1.36 (0.86–2.15) | 1.61 (0.99–2.60) |
| **Marital status** | | | | | | |
| Without couple | 1 (Reference) | 1 (Reference) | 1 (Reference) | 1 (Reference) | 1 (Reference) | 1 (Reference) |
| With couple | 0.83 (0.65–1.05) | **0.74 (0.59–0.92)** | 0.95 (0.71–1.28) | 1.03 (0.74–1.43) | 1.12 (0.68–1.84) | 1.32 (0.80–2.19) |
| **Years of diasease** | | | | | | |
| 1–10 years | 1 (Reference) | 1 (Reference) | 1 (Reference) | 1 (Reference) | 1 (Reference) | 1 (Reference) |
| ≥10 years | **1.37 (1.06–1.76)** | **1.34 (1.05–1.71)** | 0.94 (0.71–1.26) | 0.85 (0.62–1.17) | 0.93 (0.58–1.47) | 0.85 (0.53–1.34) |
| **Treatment** | | | | | | |
| No pharmacological treatment | 1 (Reference) | 1 (Reference) | 1 (Reference) | 1 (Reference) | 1 (Reference) | – |
| OAD | 2.04 (0.77–5.37) | 1.75 (0.70–4.38) | 0.97 (0.56–1.68) | 0.99 (0.58–1.71) | 1.01 (0.45–2.27) | – |
| Insulin | 2.5 (0.89–7.02) | 1.70 (0.62–4.65) | 1.25 (0.65–2.39) | 1.29 (0.65–2.56) | 0.94 (0.29–3.05) | – |
| OAD plus insulin | **3.22 (1.24–8.36)** | **2.57 (1.05–6.32)** | 1.06 (0.59–1.88) | 1.13 (0.62–2.04) | 0.75 (0.29–1.92) | – |
| **Complications** | | | | | | |
| No complications | 1 (Reference) | 1 (Reference) | 1 (Reference) | 1 (Reference) | 1 (Reference) | – |
| At least 1 complication | 0.98 (0.76–1.25) | 0.80 (0.62–1.03) | 1.13 (0.85–1.51) | 1.23 (0.89–1.69) | 0.80 (0.49–1.31) | – |
| **IPAQ** | | | | | | |
| HEPA active | 1 (Reference) | 1 (Reference) | 1 (Reference) | 1 (Reference) | 1 (Reference) | – |
| Minimally active | **1.34 (1.04–1.73)** | **1.63 (1.23–2.15)** | 0.89 (0.60–1.33) | 0.91 (0.61–1.37) | 0.86 (0.45–1.65) | – |
| Inactive | 1.12 (0.83–1.53) | 1.31 (0.98–1.75) | 0.95 (0.66–1.36) | 0.93 (0.64–1.34) | 1.19 (0.70–2.02) | – |
| **Depression** | | | | | | |
| No | 1 (Reference) | 1 (Reference) | 1 (Reference) | 1 (Reference) | 1 (Reference) | – |
| Yes | 1.05 (0.82–1.36) | 1.17 (0.94–1.45) | 1.00 (0.74–1.36) | 0.93 (0.67–1.27) | 1.54 (0.98–2.42) | – |
| **EATDM-III** | 1.00 (0.99–1.01) | 1.00 (1.00–1.02) | 1.00 (0.98–1.01) | 0.99 (0.98–1.01) | 0.98 (0.96–1.01) | 0.98 (0.96–1.00) |

**Notes.**
[a]Adjusted model for gender, marital status, years of disease, treatment, complications, IPAQ, depression and EATDM-III.
[b]Adjusted model for gender,age, marital status, years of disease and EATDM-III.
IPAQ, International Physical Activity Questionnaire; OAD, Oral antidiabetics; EATDM-III, Escala de adherencia al tratamiento de la diabetes mellitus tipo II.

## DISCUSSION

### Main findings and comparison with other studies

Only 7.5% of the patients achieved the levels of glycemic control in T2DM recommended by the ADA (glycosylated hemoglobin, LDL-cholesterol and blood pressure). In Latin America, a Colombian retrospective study with a bigger sample size got a similar result of 6.9% (*Machado-Alba, Moncada-Escobar & Gaviria, 2009*). There are several differences

between these two studies. For example, all participants in the Colombian study were from the social security health program, suggesting they had better access to care for controlling their blood glucose levels and follow up care compared to our patients. The Colombian study also had a bigger sample size from a multicenter study with a follow-up period of one year (*Machado-Alba, Moncada-Escobar & Gaviria, 2009*). In a twin cross-sectional study performed in India with one hundred participants and three time points—baseline, three months, and six months—HbA1c, blood pressure and LDL-cholesterol were evaluated (*Menon & Ahluwalia, 2015*). Only one participant (1%) achieved the ADA's guidelines at three months, but at six months this number increased to three participants achieving the recommended guidelines (3%). Due to some characteristics of our population, mostly advanced age and comorbidities, we also considered another cutoff point of HbA1c$\leq$ 8% for controlled diabetes, which indicated an increase in controlled disease from 7.5% to 14.2%. Even though there's an increase, it is still not enough for the achievement of ADA recommendations. Furthermore, 26.67% of patients had HbA1c >9%; this is untenable and incompatible with a good healthcare system and appropriate quality of attention to patients with diabetes.

Of all participants in this study, 68.33% had poor glycemic control, 37.50% had uncontrolled blood pressure and 60.83% had an abnormal lipid profile. These are worrisome results as achieving the ADA recommendations is pertinent in decreasing micro-vascular and macro-vascular complications and death in T2DM patients. This is shown in the Steno 2 study, which compared intensive treatment against conventional treatment in primary outcomes of T2DM patients (*Gaede et al., 2008*). The outcomes of this study were positive for intensive treatment having an overall-death hazard ratio of 0.54 (95% CI [0.32–0.89]; $p = 0.02$), cardiovascular-death hazard ratio of 0.43 (95% CI [0.19–0.94]; $p = 0.04$) and cardiovascular-events hazard ratio of 0.41 (95% CI [0.25–0.67]; $p < 0.001$).

In a systematic review of 14 studies that involved 19 countries: 55.5% of subjects had HbA1c > 7% (53 mmol/mol), 64.8% of subjects had blood pressure > 130/80, and 48.6% subjects had LDL-cholesterol > 100 mg/dL (*Menon & Ahluwalia, 2015*). The difference with these studies is that they were mainly conducted in high-income countries (17 out of 19 were high income countries) (*Pinchevsky et al., 2015*). In the cross-sectional study in India, they measured HbA1c, blood pressure and LDL-cholesterol at three time points (*Menon & Ahluwalia, 2015*). The percentage of participants not meeting each of the ADA recommendations at the three time points (baseline, three, and six months) were: HbA1c > 7% (53 mmol/mol) (55%, 47%, 45%), blood pressure > 130/80 (73%, 77%, 75%), and LDL-cholesterol (63%, 67%, 60%).

Interestingly, in our study, participants who were older than 65 and in a relationship were able to achieve better glycemic control. Meanwhile having more than 10 years of disease, OAD plus insulin treatment, and doing minimal physical activity was associated with poor glycemic control. There was not a significant association between the above analyzed variables and LDL-cholesterol or blood pressure.

Participants older than 65 were more likely to have good glycemic control (PR = 0.59; 95% CI [0.44–0.78]) than their younger counterparts. Similar results were found

in other studies, for example in Mexico and the United States (*Flores-Hernández et al., 2015*; *Ali et al., 2012*) , where a significant relationship between good glycemic control and an elderly population were found. Furthermore, the ACCADEMY study group (*Giorda et al., 2015*) showed that patients older than 65 achieve the glycemic goal HbA1c < 7% (53 mmol/mol) more rapidly than their younger counterparts, with older patients achieving the recommended levels within 15 months compared to 21 months of treatment.

It has also been shown that the years of disease have a significant impact towards glycemic control, which may be due to decreased insulin production from decreased beta-cell function after 5–10 years of disease (*Levy et al., 1998*). In agreement, our participants with a diagnosis of ≥10 years were more likely to have a poor glycemic control. This is supported by a study conducted in Jordan (*Al-Akour, Khader & Alaoul, 2011*) , where those who had their diagnosis 10 or more years previously had increased odds of developing poor glycemic control (OR = 1.53; 95% CI [1.09–2.17]). In summary, these results strongly suggest that duration of disease has a greater impact on glycemic control than patient age.

Another variable associated with good glycemic control was marital status. In this study, being in a relationship was shown to have protective effects with regards to the HbA1c goals and consequently better prognosis. This is reinforced by several studies where they found that having a partner is associated with better adherence to treatment (*Ali et al., 2012*; *Thompson, Auslander & White, 2001*; *Harris, National & Nutrition Examination, 2001*) and decreased odds of poor glycemic control (*Al-Akour, Khader & Alaoul, 2011*).

Participants with no pharmacological treatment had better glycemic control in comparison with OAD or OAD plus insulin. Similar results were reported in the Center for Disease Control and Prevention supplement (*Ali et al., 2012*). This can be explained because use of insulin alone or insulin in combination with oral drugs is used to treat long-standing forms of the disease, whereas management without pharmacological treatment is used for milder form of the disease (*Ali et al., 2012*). *Ahmad, Islahudin & Paraidathathu (2014)* demonstrated that all pharmacological treatment patterns (monotherapy, combination of OAD, insulin and OAD plus insulin) were more likely— in varying degrees—to have poor glycemic control. This may be associated with adverse effects and a lack of adherence to the treatment regimen.

Association of physical activity, through IPAQ score, showed that minimal physical activity was associated with poor glycemic control ( PR = 1.34 ; 95% CI [1.04–1.73]) compared to HEPA active patients. Similar results were found by Abdel et al., who found a statistically significant association between patients with uncontrolled T2DM and lack of activity (71.6% vs 28.4%, $p < 0.001$) (*Serour et al., 2007*).

According to EATDMIII scale, 56.99% patients with diabetes were adherent to non-pharmacological treatment. However, it was still beneath the rate of 66% found by *Alayón & Mosquera-Vásquez (2008)* in Colombia. It should be noted that they utilized a self-administered questionnaire SDSCA (summary of diabetes self-care activities) which does not take into account family support or community organization (*Alayón & Mosquera-Vásquez, 2008*). Both of these factors were analyzed with the EATDM III (*Villalobos-Pérez*

*et al., 2006*). In Table 2, we illustrate that the median for EATDMIII scale was similar for the patients with good glycemic control and for the patients with poor glycemic control which suggests that glycemic control and adherence might be influenced by other factors such as knowledge of the disease (*Milla, Perez & Rodriguez, 2008*), disease characteristics, environment, and intrapersonal and interpersonal factors (*WHO, 2003*).

Prevalence of depression in our patients with T2DM was 33.33%, which is five times the prevalence of depression reported in the general population (6.7%) (*Instituto Especializado de Salud Mental Honorio Delgado –Hideyo Noguchi, 2002*). However, there was not a significant correlation between depression and glycemic control in our sample. Some data tends to demonstrate a higher prevalence of depression in persons with T2DM when compared to their diabetes-free counterparts, but the evidence is inconsistent (*Hamer, Batty & Kivimaki, 2011*; *Fisher et al., 2010*). Some longitudinal studies showed that distress can be linked specifically to diabetes and its management, but neither clinical depression nor depressive symptoms seem to be associated with glycemic control (*Crispin, Robles & Bernabe, 2015*; *Hamer, Batty & Kivimaki, 2011*). Several factors including biochemical changes that are directly caused by diabetes, diabetes treatment, or distress associated with living with this disease has to be studied in order to find an association. Although we did not find an association between depression and glycemic control, this may be because the power of our sample (for this particular association) was very low (28%).

Factors significantly associated with poor glycemic control were not associated with LDL-cholesterol or blood pressure, suggesting that there are different factors that we have not explored and could be associated.

## Limitations and strengths

This study was small and localized to a single hospital in Peru. The small sample size and single site limits our generalizability and may have limited our scope of findings. Moreover, patients who participated in the study are representative of a low-income urban area, so we are unable to generalize the results to the whole population. The socio-economic characteristics of our population may have influenced rates of adherence, physical activity, and depression symptoms (*Ramos et al., 2014*; *Popkin, 2001*).

Another limitation of this study is that currently, in addition to ADA recommendations, a patient-centered approach is promoted to individualize treatment plans taking into consideration life expectancy, disease duration, complications, comorbidities, hypoglycemia, and psychological status. Unfortunately, we do not have enough information to determine the adequate HbA1c level for each participant. Despite this limitation, this study is important and highlights a challenge to improve the health system for people with T2DM to achieve a better metabolic control, keeping in mind the necessary focus on individualization of the intensity of treatment and essential security aspects. Our findings can help to direct better management of patients with T2DM. We have shown that patients are not achieving the standards recommended by the ADA and now we can try to achieve them by targeting the significant factors including: increasing physical activity, protecting relationships with psychological couple therapy, rechecking diabetes

medication, and enhancing adherence. Further studies should be conducted to determine the influence of lifestyle factors in management of T2DM.

## Conclusions

The quality of control of HbA1c, LDL-cholesterol, and blood pressure was surprisingly low in patients with T2DM in this study. In comparison to similar studies, these levels are the lowest in the literature. Additionally, patients also exhibited low glycemic control. This situation cannot solely be explained by poor adherence, but it could be explained in combination with depression or low physical activity. Further studies are needed to fully understand this problem and until this is achieved, we cannot make the necessary recommendations to address this problem.

# ACKNOWLEDGEMENTS

The authors thank Valerie Paz-Soldán and Amy Powell for their valuable comments and review onto this article, and Ray Ticse, Miguel Pinto, Mariela Florez, and Carmela Costta for their help in patient recruitment.

## Funding

This study was supported by an investigation fund (No. 02231001) given by the Faculty of Medicine "Alberto Hurtado" of the Universidad Peruana Cayetano Heredia on 2011. The funders had no role in study design, data collection and analysis, decision to publish, or preparation of the manuscript.

## Grant Disclosures

The following grant information was disclosed by the authors:
Faculty of Medicine "Alberto Hurtado": No. 02231001.

## Competing Interests

The authors declare there are no competing interests.

## Author Contributions

- Irma Elizabeth Huayanay-Espinoza, Felix Guerra-Castañon and María Lazo-Porras conceived and designed the experiments, performed the experiments, analyzed the data, contributed reagents/materials/analysis tools, wrote the paper, prepared figures and/or tables, reviewed drafts of the paper.
- Ana Castaneda-Guarderas, Nimmy Josephine Thomas, Ana-Lucia Garcia-Guarniz, Augusto A. Valdivia-Bustamante and Germán Málaga conceived and designed the experiments, performed the experiments, analyzed the data, contributed reagents/materials/analysis tools, wrote the paper, reviewed drafts of the paper.

## Human Ethics

The following information was supplied relating to ethical approvals (i.e., approving body and any reference numbers):

The study protocol was approved by the Ethical Review Board of the Universidad Peruana Cayetano Heredia and Ethical Committee of Hospital Cayetano Heredia.

## Data Availability

The raw data has been supplied as Supplemental Dataset.

## Supplemental Information

Supplemental information for this article can be found online at http://dx.doi.org/10.7717/peerj.2577#supplemental-information.

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
