# Peer review of "Metabolic control in patients with type 2 diabetes mellitus in a public hospital in Peru: a cross-sectional study in a low-middle income country"

_PeerJ, doi:10.7717/peerj.2577_

## Round 0.1 · original submission · Major Revisions

The study by Huayanay-Espinoza et al. is of potential interest for the readers of PeerJ. However the novelty of the study is not clear.
hence, the Authors should clearly discuss what this study adds to the knowledge.

Moreover, there are some methodologic issues that should better addressed, as discussed in detail by Reviewers 2 and 3.

Some minor changes are also needed, as recommended by Reviewer 1.

·

Basic reporting

Espinoza and colleagues aimed to investigate the quality of type 2 diabetes mellitus (T2DM) management according to the American Diabete Association (ADA) recommendations. The second object of this study was to evaluate if demographic characteristics and lifestyle choices could influence the metabolic control of the T2DM. To reach these aims the authors and enrolled 123 T2DM patients from a public hospital in Lima, Peru. They showed that only the 4.1% of the patients achieved the metabolic control according to ADA recommendations. Therefore, they demonstrated that the marital status as the no pharmacological treatment were associated with good glycemic control.

Experimental design

The authors conducted a cross-sectional study of T2DM patients. All the inclusion and exclusion criteria were well explained. The authors also elucidated the definition of poor metabolic control according to the ADA recommendations as the used tests (ADTDM-III to measure the adherence and the CES-D to evaluate the depression).

Validity of the findings

Statistical analysis were well conducted. The authors validated their results with a multivariable analysis adjusted for several confounding factors. All the results are well explain ad discussed.

Additional comments

The paper is well written, however few correction are necessary:
1) The number of the reference in the text could be before the comma or the full stop. i.e. line 49: “respectively [2].However…”. Line 50: “…the last seven years [3].” Please, correct it in all the manuscript.
2) Line 133, add a reference “…has previously been used in Costa Rica []”.
3) Results: all the results could be written with the units ± SD. You don’t have to write SD between the parenthesis. i.e. line 142: correct as “61.84 years ± 11.10”. Line 143-144: correct as “LDL was 107.61 mg/dl ± 39.66”. Please provide to correct it in the entire paragraph.
4) Table 1. Delete the parenthesis when you write the SD. i.e.”Age, mean ± SD” and the results will be “61.84±11.10”. To the contrary, you have to add “(%)” in the variables column for each variables where you use it. i.e. “Age group (%)” ore “Gender (%)”. Please, for blood pressure correct as “blood pressure (mmHg), mean (%)” because you do not used the SD for that variables.
5) Table 2. Add for each variable “±SD” or “(%)”. i.e. “Age±SD”, “Age per range (%)”.
6) Table 3. Correct a type mistake in the legend “adjusted” instead of “adjused”.

Reviewer 2 ·

Basic reporting

This article attempts to analyze how are controlled patients with type 2 diabetic according to the latest guidelines. It is a topic that despite being widely studied, serves notice to readers of the magazine to try to make things better. The structure of the submitted article is correct acording to the guide of the journal. The article is written in a not very professional basic English. Table 1 and 2 could be presented as a single table.

Experimental design

It is a cross-sectional study in a small cohort. Indeed, the latest guides talk about the individualization of treatment in diabetic patients. So the criteria for defining poor metabolic control, it does not seem quite right

Validity of the findings

The conclusions are appropriately stated, are connected to the original question investigated, but the results are not novel

Reviewer 3 ·

Basic reporting

No Comments

Experimental design

No Comments

Validity of the findings

No Comments

Additional comments

Huayanay-Espinoza et al. report an observational cross-sectional study where they sough to assess the patients’ achievement of ADA guideline recommendations in a developing country (Peru). Their study is interesting; the research question is clearly defined, given its scope fits well in this journal.

I find, the following areas for closer scrutiny and clarification:

1. The authors follow the recommendation of the ADA of glycemic control. However, the HbA1c of 7% is not a “one-size-fits-all” recommendation and now individualization of HbA1c target should be made by taking into consideration multiple factors such as co-morbidities, life expectancy, risk or episodes of hypoglycemia etc. Not sure that the claim they make of poor glycemic control can be generalized.

2. A mean HbA1c of 8.23% for a population over 60 years in which 50% have 10 years of diagnosis does not look bad at all. Will ask the authors what is there a real difference of tight glycemic control (HbA1c <7.0%) over patient-important micro- (end-stage renal disease, dyalisis, blindness, clinical neuropathy) and macrovascular outcomes (mortality, non-fatal MI, stroke) when compared to conventional glycemic control- do all patients will be better of if they were to get to a HbA1c of <7.0% (HbA1c 7.0-8.5%) (Montori et al 2009 Annals Internal Medicine, Boussageon et al BMJ 2009, Heimmingen Cochrane SR, 2013)

3. A very important unintended consequence of tight glycemic control is hypoglycemia- which impairs quality of life and it is related to adverse cardiovascular outcomes included death. Did the authors look for this? If so can they report it? Can we really say we are delivering high quality care when we put a patient who has an HbA1c of 7.3% on insulin or another medication just because he or she is not meeting the HbA1c goal of <7.0%. Is it really worth the burden of treatment, the out-of-pocket cost, the risk of adverse outcomes, and even more important what is the real benefit the individual patient will have?

4. Can the author comment why did they choose to “poor metabolic controlled LDL-C ≤100 mg/dl” if after the AHA/ACC 2013 Guidelines the ADA advocates not for a LDL-C goal, but rather to the use of moderate to high dose of statin therapy depending on the 10-year cardiovascular risk and the patients age?

5. An important limitation is that they generalize the group as a whole with out considering comorbidities, time of diabetes diagnosis, medications (use of insulin) etc. For instance they claim “that patients were not on a pharmacological treatment had better glycemic control” is obvious as this patients probably are at an earlier stage of the disease, probably are younger, with less comorbidities, less burden of treatment, less burden of illness etc. The goals in this population cannot be the same as in the other side of the spectrum.

---

## Round 0.2 · accepted · Accept

The Reviewer who suggested you to extensively revise your manuscript appreciated your work, and found the new version considerably improved.

Reviewer 3 ·

Basic reporting

No Comments

Experimental design

No Comments

Validity of the findings

No Comments

Additional comments

The authors have worked hard in this new version. My concerns have been responded fairly.